# Effects of *in utero* delta-9-tetrahydrocannabinol (THC) exposure on fetal and infant musculoskeletal development in a preclinical nonhuman primate model

Samantha A. Moellmer[1], Olivia L. Hagen[2], Parsa A. Farhang[3], Victoria R. Duke[1], Meghan E. Fallon[4], Monica T. Hinds[1], Owen J. T. McCarty[1], Jamie O. Lo[2,5☯], Karina H. Nakayama[1☯]*

1 Department of Biomedical Engineering, Oregon Health & Science University, Portland, OR, United States of America, 2 Division of Reproduction and Developmental Sciences, Oregon National Primate Research Center, Oregon Health & Science University, Beaverton, OR, United States of America, 3 Department of Molecular Microbiology and Immunology, Johns Hopkins University, Baltimore, MD, United States of America, 4 Yale Cardiovascular Research Center, Section of Cardiovascular Medicine, Department of Internal Medicine Yale School of Medicine, New Haven, CT, United States of America, 5 Department of Obstetrics and Gynecology, Division of Maternal Fetal Medicine, Oregon Health & Science University, Portland, OR, United States of America

☯ These authors contributed equally to this work.
* nakayaka@ohsu.edu

**Data Availability Statement:** Nanostring nCounter raw and normalized data has been made available on Kaggle, doi: 10.34740/kaggle/dsv/7853952.

## Abstract

The endocannabinoid system (ECS) plays a major role in the maintenance of bodily homeostasis and adaptive response to external insults. It has been shown to regulate crucial physiological processes and behaviors, spanning nervous functions, anxiety, cognition, and pain sensation. Due to this broad activity, the ECS has been explored as a potential therapeutic target in the treatment of select diseases. However, until there is a more comprehensive understanding of how ECS activation by exogenous and endogenous ligands manifests across disparate tissues and cells, discretion should be exercised. Previous work has investigated how endogenous cannabinoid signaling impacts skeletal muscle development and differentiation. However, the effects of activation of the ECS by delta-9-tetrahydrocannabinol (THC, the most psychoactive component of cannabis) on skeletal muscle development, particularly *in utero*, remain unclear. To address this research gap, we used a highly translational non-human primate model to examine the potential impact of chronic prenatal THC exposure on fetal and infant musculoskeletal development. RNA was isolated from the skeletal muscle and analyzed for differential gene expression using a Nanostring nCounter neuroinflammatory panel comprised of 770 genes. Histomorphological evaluation of muscle morphology and composition was also performed. Our findings suggest that while prenatal THC exposure had narrow overall effects on fetal and infant muscle development, the greatest impacts were observed within pathways related to inflammation and cytokine signaling, which suggest the potential for tissue damage and atrophy. This pilot study establishes feasibility to evaluate neuroinflammation due to prenatal THC exposure and provides rationale

**Funding:** The sources of funding that supported this works are as follows: NIH/NIAMS R01AR080150 (KN), NIH/NIAMS R01HL101972 (OM), NIH/NICHD R01HD097367 (JL), NIH/NIDA DP1 DA056493 (JL), NIH/OD P51-OD-011092 (JL), March of Dimes (JL), Silver Family Innovation Award (JL), Oregon Health & Science University SEED Grant (JL), and the Society for Maternal–Fetal Medicine (JL). The funders had no role in study design, data collection and analysis, decision to publish, or preparation of the manuscript.

**Competing interests:** The authors have declared that no competing interests exist.

for follow-on studies that explore the longer-term implications and functional consequences encountered by offspring as they continue to mature.

## 1. Introduction

The endocannabinoid system (ECS) is a network of neuromodulatory pathways that regulates central nervous system development, synaptic plasticity, and the body's response to endogenous and environmental insults. The ECS is comprised of 3 main constituents: endogenous cannabinoids (endocannabinoids), endocannabinoid receptors, and the enzymes responsible for the synthesis and degradation of the endocannabinoids [1]. The most studied receptors in the ECS are two G-protein-coupled receptors, known as the cannabinoid type-1 and type-2 receptors (CB1R and CB2R, respectively). CB1R is the most abundant cannabinoid receptor and is widely expressed in the central nervous system [1]. There is conflicting evidence surrounding CB2R expression in the central nervous system, with some studies suggesting CB2R is weakly expressed in hippocampal and brain stem neurons under normal physiological conditions [2, 3]. However, under inflammatory conditions or injury, CB2R is highly expressed on reactive microglia [4–6]. Outside of the nervous system, CB2R is highly expressed in peripheral immunological tissues [4]. The most well-characterized endocannabinoids are 2-arachidonoyl glycerol (2-AG) and arachidonoyl ethanolamide (AEA), and while they are structurally similar, they participate in distinct biological processes [7]. While the activity of 2-AG is agonistic to both CB1R and CB2R with low affinity, AEA only interacts with CB1R with high affinity [8]. Much work has been done to characterize CB1R signaling at the synapse [9] and studies have shown that CB1R expression patterns vary across a wide variety of non-nervous system tissues, such as skeletal muscle, liver, and pancreatic islets [10]. However, ECS signaling and subsequent outcomes in these tissues are not well defined.

Previous work has shown that not only is CB1R expressed in skeletal muscle, but signaling through the receptor modulates insulin sensitivity and is a key player in skeletal muscle energy homeostasis [11]. Adding to this, 2-AG signaling through CB1R has led to repression of myoblast differentiation into mature myotubes [12]. Much of the previous research looking at CB1 signaling in skeletal muscle has focused on activation by endogenous cannabinoids. However, exogenous cannabinoids, such as delta-9-tetrahydrocannabinol (THC), can readily interact with highly vascularized tissues, such as skeletal muscle, following circulation in the blood stream.

THC is the main psychoactive component in cannabis and acts as a partial agonist on CB1R and CB2R [13–15]. In addition to its role as a partial agonist on the CB2R receptor, THC can inhibit the activity of cannabinoid ligands on CB2R, thereby also acting as a weak antagonist in select settings [16]. Modulation of the ECS has received attention for the therapeutic potential in the treatment of pain, inflammation, metabolic disorders, and neurodegenerative disease [17]. Notably, the effect of THC on neuroinflammation in disease settings has led to improved health outcomes. Specifically, administration of THC prevented the clinical development of experimental autoimmune encephalomyelitis (a widely used animal model of multiple sclerosis) and led to reduction of inflammation in nervous system tissue [18]. However, in the setting of systemic inflammation or obesity, administration of THC alone produced no change or an increase in inflammatory cytokines, respectively [19, 20]. These results suggest that the therapeutic benefit of THC heavily depends on the (patho)physiological setting.

Considering that increased activation of CB1R through 2-AG can disrupt myoblast differentiation, it is important to consider how increased activation through exogenous

endocannabinoids might affect early skeletal muscle development. To address this research gap, we leveraged a translational non-human primate model to determine the impact of chronic THC exposure on fetal and infant musculoskeletal development. In this model, female adult rhesus macaques received daily THC edibles pre-conception and continued throughout pregnancy. THC can readily cross the placenta [21] and bind to endocannabinoid receptors present in the placenta and fetal organs, including skeletal muscle [22, 23]. This model enables the study of prenatal endocannabinoid receptor activation on early skeletal muscle development in the offspring. The objective of this pilot study was to characterize the neuroinflammatory response from the stimulation of endocannabinoid signaling on fetal and infant musculoskeletal development. Towards this aim, control and THC-exposed skeletal muscle, at two stages of offspring development (fetal and infant), were collected and evaluated by histomorphology and for differential gene expression using a Nanostring nCounter neuroinflammatory panel. These initial findings are intended to shed light on key pathways impacted by stimulation of the ECS during early development.

## 2. Materials and methods

### 2.1 Experimental design

All protocols, including THC administration, were approved by the Institutional Animal Care and Use Committee (IP0001389) at the Oregon National Primate Research Center (ONPRC). Methods are in accordance with the ARRIVE guidelines [24]. Animals were observed at least twice a day by trained animal care technicians and every effort is made by both veterinary and research staff to minimize pain and distress in the rhesus macaques as clinically needed. During the course of the study, clinical signs of distress in the pregnant dams such as weight loss, anorexia, cachexia, self-injury, or failure to thrive were assessed by ONPRC Division of Comparative staff and the ONPRC Behavioral Management Program to ensure psychological health and well-being. When possible, positive reinforcement training was used to train rhesus macaques to cooperate with various procedures, thus reducing the stress associated with those procedures. Rhesus macaques were housed in full contact pairs with a compatible social partner for the duration of the study in accordance with the Guide for the Care and Use of Laboratory Animals [25]. This study also included cookies containing THC (THC edible) that were made using research-grade THC obtained directly from the National Institute on Drug Abuse (NIDA) Drug Supply Program [26, 27].

This study used indoor-housed adult, female rhesus macaques (n = 18) of similar size (~6–7 kg) maintained on a standard chow diet (TestDiet, St. Louis, Missouri) given twice a day in addition to fresh produce or other food enrichment daily and tap water available ad libitum. The animals were randomly divided into two groups whose offspring were studied either at the fetal (n = 10) or infant (n = 8) timepoint. The adult females in each cohort were further divided by random assignment to control or THC-exposed groups. All cohorts underwent time-mated breeding. For the THC-exposed group, THC dose was increased up to 0.36 mg/kg/day following published medical cannabis acclimation recommendations [28] four months prior to time-mated breeding. The specific THC dose received by the THC-exposed groups prior to time-mated breeding was as follows: weeks 0–3 animals received a dose of 0.07 mg/kg/day, weeks 4–6 animals received a dose of 0.14 mg/kg/day, weeks 7–9 animals received a dose of 0.29 mg/kg/day, and weeks 10–12 animals received a dose of 0.36 mg/kg/day. Edibles were administered in the morning prior to daily chow with confirmation of complete ingestion. THC-exposed animals in both cohorts were maintained at a dose of 0.36 mg/kg/day from before time-mated breeding until offspring necropsy (Fig 1). All methods of euthanasia utilized at the ONPRC are consistent with the American Veterinary Medical Association

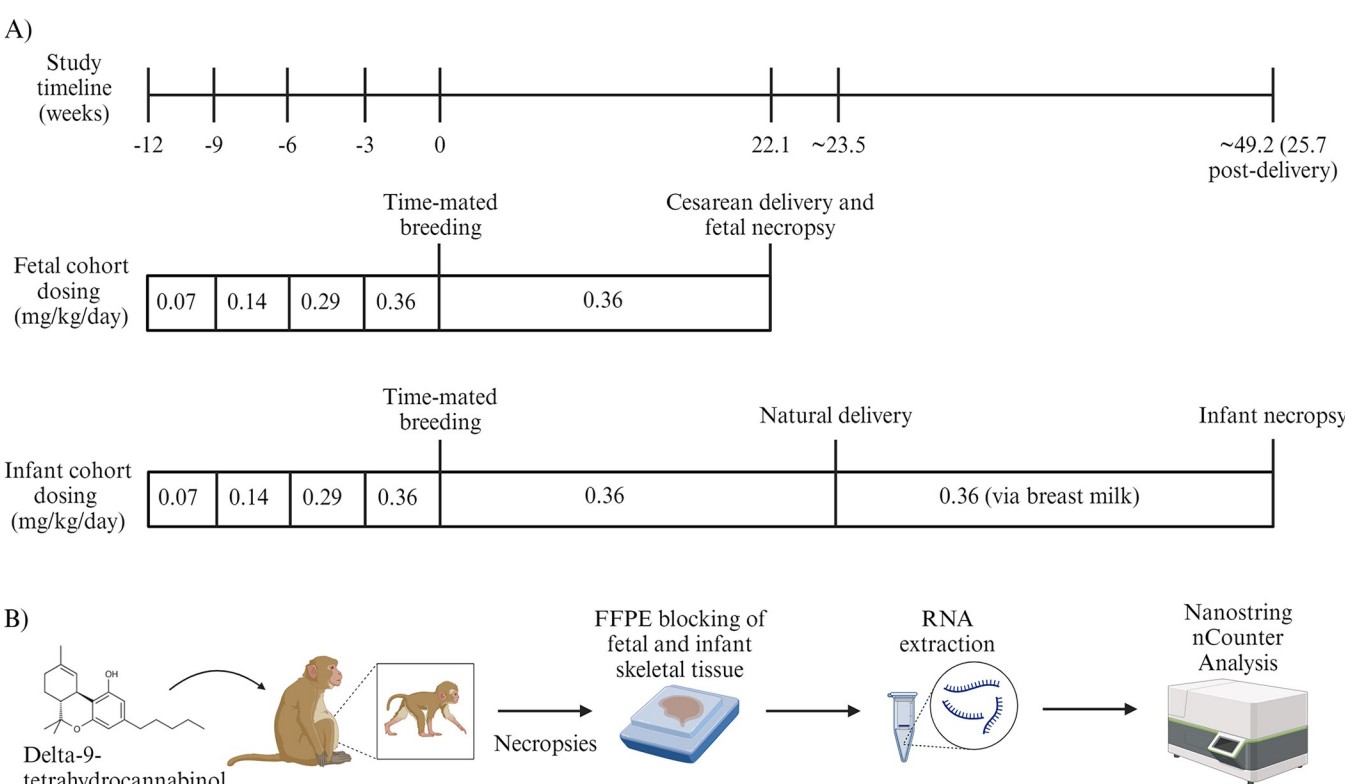

**Fig 1. Experimental design and differential gene expression workflow.** A) Delta-9-tetrahydrocannabinol dosing timeline for fetal and infant rhesus macaque cohorts. B) RNA sample processing and differential gene expression workflow using NanoString nCounter. Created with BioRender.com.

Guidelines for the Euthanasia of Animals [29], sodium pentobarbital was delivered intravenously followed by exsanguination under a deep surgical plane of anesthesia.

The fetal cohort offspring were delivered via cesarean section and underwent necropsy on gestational day 155. The infant cohort offspring were delivered naturally at full term (~165 days) and underwent necropsy 180 days post-delivery. The offspring in the fetal cohort included 5 females and 5 males. The offspring in the infant cohort included 6 females and 2 males. Skeletal muscle was isolated from the rectus femoris of both fetal and infant rhesus macaques and processed for storage in formalin-fixed, paraffin-embedded blocks (FFPE). Blocks were stored at 4˚ C until used for sectioning.

## 2.2 RNA isolation and evaluation

RNA was extracted from rectus femoris FFPE tissues for the analysis of gene expression. Total RNA extraction from fetal and infant rhesus macaques was performed using AllPrep DNA/RNA FFPE Kit (Qiagen, catalog #80234). Sample input did not exceed three 15 µm thick sections of 150 mm$^2$ surface area, which was optimized per individual sample. RNA isolation protocol was performed in accordance with manufacturer recommendations, including two additional buffer wash steps to increase RNA purity. Following isolation, overnight ethanol precipitation was performed to de-salt RNA and increase concentration. Briefly, 0.1 volume of sodium acetate and 3 volumes of 100% ethanol were added to RNA and precipitated at -20˚C overnight. RNA was pelleted through centrifugation and washed twice with 75% ethanol. Following the final wash, RNA was allowed to air dry for 10 minutes and resuspended in nuclease-free water.

RNA purity and concentration were evaluated using a NanoDrop spectrophotometer. RNA integrity (RIN) and quality (DV200) were assessed using Agilent 2100 Bioanalyzer and RNA 6000 Nano Chip. Samples selected for differential gene expression analysis had A260/280 value $\geq$ 1.5, A260/230 value $\geq$ 0.67, and concentration $\geq$ 30 ng/μL. Analyzed samples had RIN values ranging between 1.1 to 2.5 and DV200 values ranging between 12–63%. The RIN values of analyzed samples are comparable and expected for FFPE tissues [30].

## 2.3 NanoString and gene enrichment analysis

The NanoString neuroinflammatory panel (catalog #115000230) comprised of 770 genes, including 13 internal reference genes for data normalization, was used to obtain gene expression data. NanoString technology was purposefully chosen for our study because this platform has been optimized for obtaining high quality readouts from RNA isolated from FFPE samples that are comparable to expression data from freshly frozen tissues [30, 31]. Each sample included an input of 150 ng of FFPE-derived RNA. Gene targets were hybridized with gene-specific barcodes using NanoString CodeSet chemistry. Hybridized barcode count was visualized on the nCounter SPRINT profiler and data was analyzed using ROSALIND. Hybridization effectiveness was assessed in ROSALIND quality control which evaluated imaging quality, binding density, negative and positive control counts, and housekeeping gene counts. Specifically, the imaging quality required greater than 0.75% field of view captured. The concentration of barcodes was required to range between 0.1 and 2.24 spots per square micron. The positive control correlation analysis was required to be greater than 0.95. The limit of detection was assessed by confirming that the counts of the positive control were higher than two times the standard deviation above the mean of the negative control. One infant cohort sample failed to meet the ROSALIND quality control criteria and was excluded from analysis. Data normalization followed NanoString nCounter Advanced Analysis protocol by dividing gene counts within a lane by the geometric mean of normalization probes from the same lane. Differential gene expression was assessed using a generalized linear model, assuming a negative binomial distribution. The model calculated fold-change and p-value for each gene. For a gene to be considered differentially expressed, a fold change $\geq$ 1.5 and p-value $\leq$ 0.05 were required. Principal component analysis (PCA) was run in R-Studio (version 2023.9.0.463) using normalized expression values to identify the axes of maximum variation among samples as well as the most influential genes contributing to differences between treatment groups. The STRING pathway and Enrichr were used to perform gene enrichment analysis [32–35]. To run this analysis, genes differentially expressed due to THC exposure were inputted into STRING and Enrichr to highlight signaling pathways.

## 2.4 Histology

FFPE rectus femoris tissues were sectioned along the transverse axis (10 micron) onto glass slides. Sections were stained with Masson-Goldner Trichrome (Millipore Sigma, catalog #1004850001) as per manufacturer recommendations. For Masson-Goldner staining, sectioned tissues were deparaffinized in SafeClear (Fisher Scientific, catalog #23–044192) and rehydrated in decreasing concentrations of ethanol. Slides were stained with Bouin's solution (Sigma Aldrich, catalog #HT10132-1L) to increase staining of muscle fibers. Slides were stained with Weigert's iron hematoxylin staining solution, rinsed with water, followed by alternating reagents (Azophloxine solution, Tungstophosphoric acid orange G solution, Light green SF solution) and 1% acetic acid (Sigma Aldrich, catalog #1.00485). Slides were treated with increasing concentrations of ethanol followed by xylene.

The density of tissue collagen was quantified for the Masson-Goldner stains using a method published by Chen et al [36]. Briefly, slides were scanned and the background was subtracted using MatLab version 23.1 (threshold = 0.8). Background subtracted RBG images were then imported into ImageJ and were deconvolved using the color deconvolution plugin [37]. Images were split into red, blue, and green components. Quantification of the red component was performed adjusting threshold to select for collagen and for total tissue. Raw pixel density was measured from binary, segmented images. Percentage of collagen within each histological section was quantified by dividing the pixel density for segmented collagen by the pixel density of the whole tissue.

## 2.5 Statistical analysis

Statistical analysis for gene expression data is described in Section 2.3. For histological analysis, data normality was confirmed through Shapiro-Wilk test and statistical differences between control and THC-exposed was evaluated by unpaired t-test. A p-value < 0.05 was considered significant. Statistical analysis was performed using GraphPad Prism 10.

# 3. Results

## 3.1 *In utero* THC exposure leads to differences in gene expression at fetal and infant developmental stages

THC exposure is associated with gene expression changes in the brain and immune cells [38, 39]. We therefore sought to evaluate the effects of prenatal THC exposure on skeletal muscle gene expression. RNA was isolated from the fetal and infant rectus femoris muscle and subsequently analyzed for differential gene expression using the NanoString nCounter platform. The samples analyzed had A260/280 between 1.5–1.9, A260/230 between 0.67–1.9, and concentration between 30–200 ng/μL. Sample RIN ranged from 1.1–2.5, and DV200 was 12–63%. In the fetal cohort, 13 differentially expressed genes (DEG) were identified (fold change $\geq$ 1.5, p-value $\leq$ 0.05). 6 genes were significantly upregulated due to THC-exposure, including *BAD*, *HMOX1*, *KDM1B*, *VEGFA*, *RELN*, and *BCL2L11*. Additionally, 7 genes were downregulated due to THC-exposure, including *TMEM144*, *HPGDS*, *P2RY12*, *PCNA*, *SLC6A1*, *NRM*, *NTHL* (Table 1 and Fig 2A). Due to the high dimensionality of this data, PCA was run to obtain

**Table 1. DEG in the fetal cohort.**

| Differentially expressed gene | Fold change | p-value |
| --- | --- | --- |
| *HMOX1* | 2.60 | 0.05 |
| *P2RY12* | -2.48 | 0.04 |
| *BCL2L11* | 2.13 | 0.04 |
| *SLC6A1* | -1.83 | $5.22e^{-3}$ |
| *NRM* | -1.83 | 0.02 |
| *RELN* | 1.82 | 0.03 |
| *VEGFA* | 1.82 | 0.05 |
| *PCNA* | -1.81 | 0.01 |
| *NTHL1* | -1.81 | 0.04 |
| *TMEM144* | -1.80 | 0.04 |
| *KDM1B* | 1.68 | 0.04 |
| *HPGDS* | -1.61 | 0.04 |
| *BAD* | 1.52 | 0.04 |

DEG in the fetal cohort and respective fold-changes and p-values.

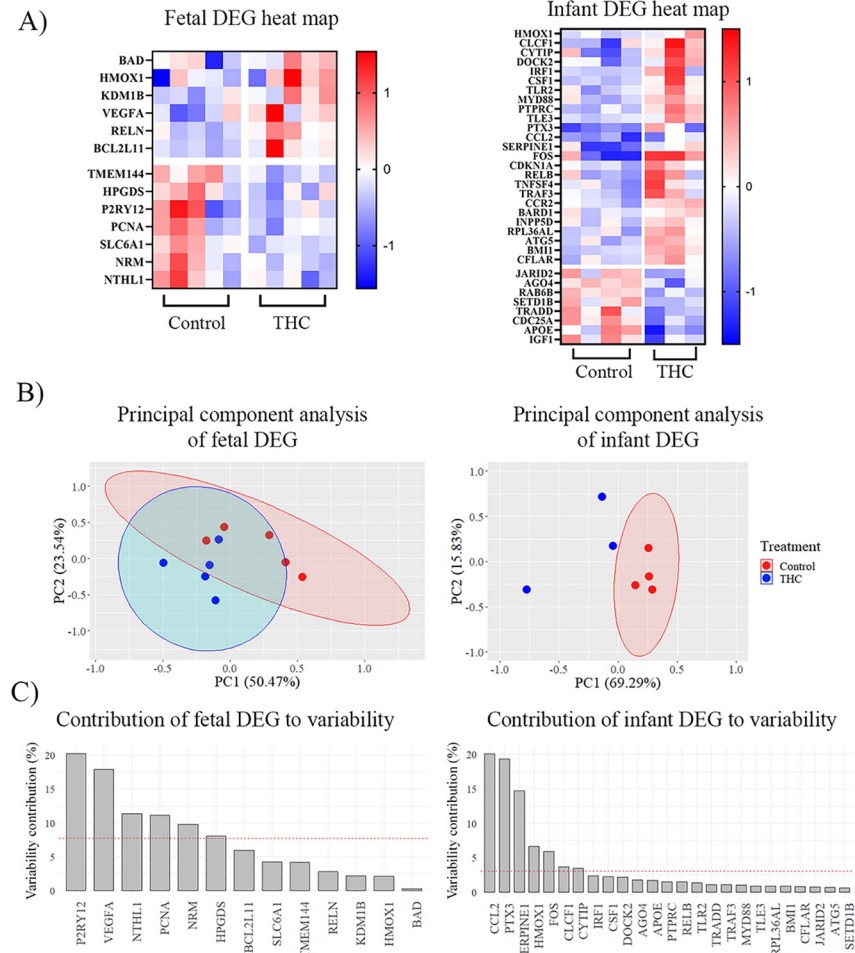

**Fig 2. *In utero* THC-exposure shifts gene expression at the fetal and infant timepoints.** A) Heatmaps of DEG between control and THC-exposed offspring in the fetal (left) and infant (right) cohorts. Values were normalized in ROSALIND by dividing gene counts within a lane by the geometric mean of normalization probes from the same lane. Differential gene expression was assessed using a generalized linear model, assuming a negative binomial distribution. B) PCA plots of control (red) vs THC-exposed (blue) differential gene expression. C) Contributions of DEG to differences between control and THC-exposed groups. The red line denotes variability contribution percentage if each gene contributed equally.

principal components (PC1 and PC2), which together accounted for 74.0% of the variation in the dataset. The PCA also identified that the genes that contributed the most to variability between the control and THC-exposed groups were *P2RY12*, *VEGFA*, *NTHL1*, *PCNA*, *NRM*, and *HPGDS* (Fig 2B and 2C).

In the infant cohort, 33 genes were differentially expressed. 25 genes were upregulated due to THC-exposure, including *CCL2*, *PTX3*, *SERPINE1*, *HMOX1*, *CYTIP*, *FOS*, *CLCF1*, *PTPRC*, *DOCK2*, *CCR2*, *TNFSF4*, *RELB*, *IRF1*, *TLR2*, *CSF1*, *TRAF3*, *INPP5D*, *BARD1*, *BMI1*, *CDKN1A*, *MYD88*, *ATG5*, *RPL36AL*, *TLE3*, and *CFLAR*. Additionally, 8 genes were downregulated due to THC-exposure, including *APOE*, *TRADD*, *CDC25A*, *IGF1*, *AGO4*, *RAB6B*, *SETD1B*, and *JARID2* (Table 2). PCA was performed to obtain principal components, which together accounted for 85.12% of the variation in the dataset. The PCA also identified that the genes that contributed the most to variability between control and THC-exposed groups were *CCL2*,

**Table 2. DEG in the infant cohort.**

| Differentially expressed gene | Fold change | p-value |
|---|---|---|
| CCL2 | 14.86 | 0.02 |
| PTX3 | 9.89 | 0.02 |
| SERPINE1 | 7.83 | 0.02 |
| HMOX1 | 3.96 | 0.02 |
| CYTIP | 3.68 | 0.02 |
| APOE | -3.22 | 0.01 |
| FOS | 3.15 | 0.02 |
| CLCF1 | 2.89 | 0.04 |
| PTPRC | 2.75 | 0.01 |
| TRADD | -2.73 | 0.05 |
| DOCK2 | 2.63 | 0.02 |
| CCR2 | 2.56 | $9.28e^{-3}$ |
| TNFSF4 | 2.56 | 0.04 |
| RELB | 2.25 | 0.04 |
| IRF1 | 2.17 | 0.05 |
| TLR2 | 2.14 | 0.03 |
| CSF1 | 2.06 | 0.03 |
| TRAF3 | 1.99 | 0.03 |
| CDC25A | -1.93 | 0.01 |
| INPP5D | 1.77 | 0.02 |
| IGF1 | -1.76 | 0.04 |
| BARD1 | 1.76 | $2.13e^{-3}$ |
| BMI1 | 1.67 | $3.52e^{-3}$ |
| CDKN1A | 1.64 | 0.02 |
| MYD88 | 1.59 | 0.01 |
| AGO4 | -1.58 | 0.02 |
| RAB6B | -1.58 | 0.01 |
| ATG5 | 1.58 | 0.02 |
| RPL36AL | 1.57 | 0.03 |
| SETD1B | -1.57 | 0.02 |
| JARID2 | -1.55 | 0.05 |
| TLE3 | 1.52 | 0.02 |
| CFLAR | 1.51 | 0.03 |

DEG in the infant cohort and respective fold-changes and p-values.

*PTX3*, *SERPINE1*, *HMOX1*, *FOS*, *CLCF1*, and *CYTIP* (Fig 2B and 2C). Taken together, these findings demonstrate that prenatal THC exposure alters gene expression profiles in the rectus femoris at the fetal and infant developmental stages.

## 3.2 DEG enriched in pathways related to inflammation and immune response

Next, we sought to characterize the interactome of the DEG, in order to describe the functional consequences of prenatal THC exposure on skeletal muscle. The STRING database was utilized to perform gene set enrichment analysis to highlight pathways dysregulated due to THC exposure (Fig 3A). In the fetal cohort, 4 of the upregulated genes, *VEGFA*, *BAD*, *BCL2L11*, and

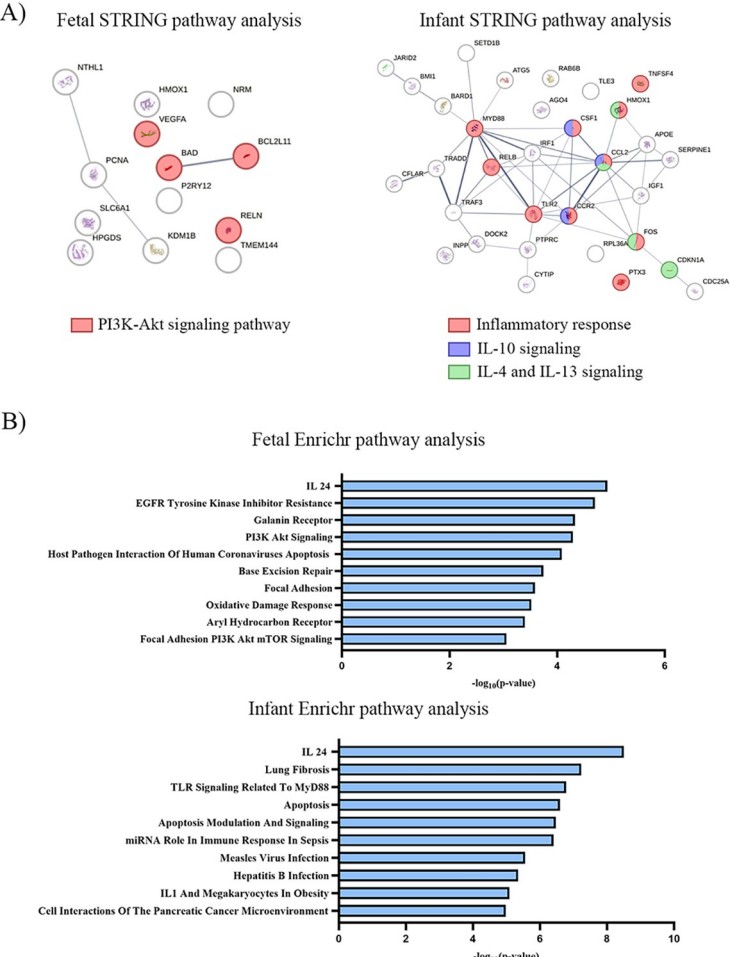

**Fig 3. Gene enrichment analysis using STRING pathway and Enrichr.** A) Gene enrichment analysis using STRING to highlight the pathways most affected by *in utero* THC exposure in fetal (left) and infant (right) cohorts. Fetal DEG involved in PI3k-Akt signaling pathway are highlighted in red. Infant DEG involved in inflammatory response, IL-10 signaling, and IL-4 and IL-13 signaling are highlighted in red, blue, and green, respectively. B) Gene enrichment analysis using Enrichr. Bar charts of the top 10 enriched terms using the WikiPathway 2023 Human gene set library for fetal (top) and infant (bottom) cohorts.

*RELN*, were found to be enriched in the PI3K-Akt signaling pathway, with a strength of 1.31. Enrichment strength is calculated by taking the logarithm of the ratio between 1) the number of DEG in the fetal cohort involved in PI3K-Akt signaling and 2) the number of proteins expected to be involved in PI3K-Akt signaling in a random protein network of the same size. In the infant cohort, 10 of the upregulated genes, *MYD88*, *RELB*, *CSF1*, *CCL2*, *CCR2*, *TLR2*, *FOS*, *HMOX1*, *TNFSF4*, *PTX3*, were enriched in the inflammatory response, with a strength of 1.16. More specifically, 4 of the upregulated genes were involved in IL-4 and IL-13 signaling, while 3 upregulated genes are involved in IL-10 signaling, with strengths of 1.39 and 1.7, respectively. All false discovery rates were ≤ 0.05, with the p-value corrected for multiple testing using Benjamini-Hochberg procedure.

Enrichr was also used to perform gene set enrichment analysis. Pathways enriched with the fetal DEG included IL-24 signaling pathway and PI3K/Akt signaling pathway. Pathways enriched with the infant DEG included IL-24 signaling and apoptosis (Fig 3B).

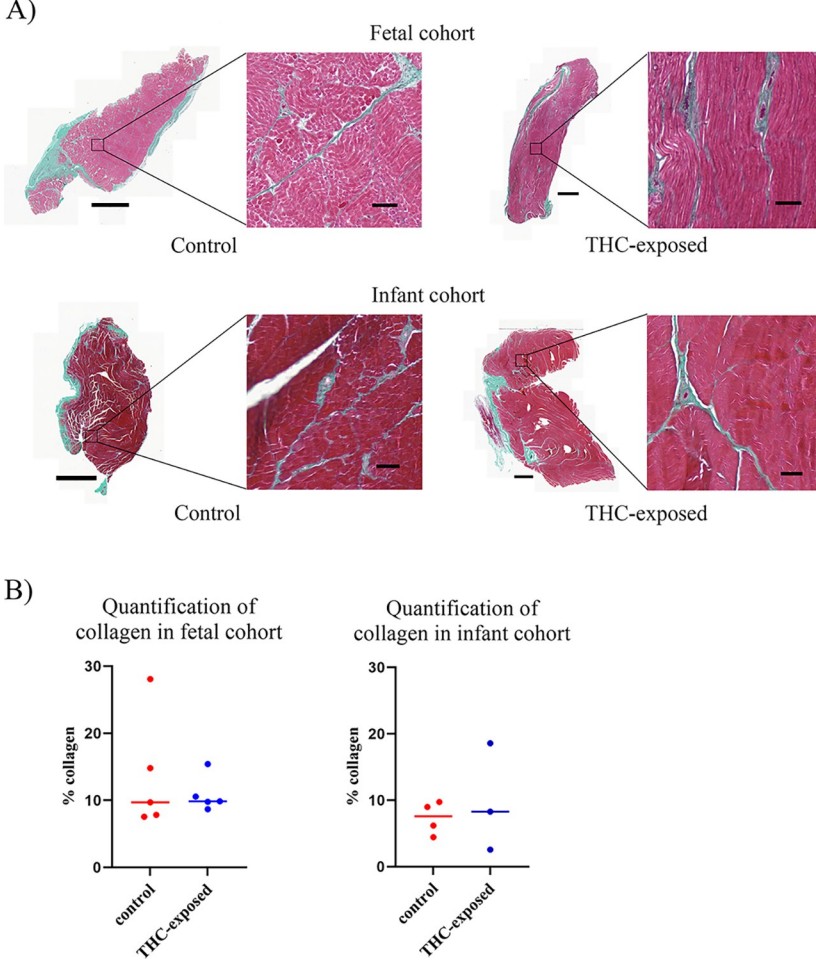

**Fig 4. Chromatic staining of fetal and infant rhesus macaque rectus femoris tissue.** A) Representative images of Masson-Goldner Trichrome staining of fetal (top) and infant (bottom) rectus femoris tissues. Whole tissue scale bars measure 2000 μm, zoomed image scale bars measure 100 μm. B) Quantification of collagen in Masson-Goldner Trichrome stain in fetal and infant cohorts. Horizontal lines represent the mean of each group. Statistical analysis was performed using unpaired t-test on GraphPad Prism 10.

### 3.3 Prenatal THC exposure did not lead to histological differences in rectus femoris collagen production

The anti-inflammatory potential of cannabis and THC has recently been explored in the prevention and reversal of fibrotic tissue [40]. To evaluate the potential development of fibrosis, we therefore quantified the effects of prenatal THC exposure on the amount of collagen present in the skeletal muscle. Sectioned FFPE tissue was stained with Masson-Goldner Trichrome and the percentage of collagen area within each tissue section was similar between control and treatment groups ($p > 0.05$, unpaired t-test, Fig 4).

## 4. Discussion

Since the discovery of ECS receptors and ligands, the pharmacotherapeutic potential of this system has been exploited. Activation of the ECS through the use of medical cannabis has led to improved outcomes in select pathologies; however, the impact of chronic, *in utero* THC exposure on offspring tissue development remains unclear. In the United States, the

prevalence of medical and recreational cannabis use is rising, including amongst individuals of reproductive age [41]. There is limited research available on the risks and long-term consequences of prenatal THC exposure on fetal and infant development. The paucity of research is partly due to limited sample sizes, patient self-reporting, different modes of cannabis delivery, varying doses of cannabis products, and confounding variables such as polysubstance use. To overcome these limitations, we used controlled administration of THC edibles to pregnant female rhesus macaques to study the effects of chronic endocannabinoid system activation on offspring skeletal muscle development.

Chronic prenatal THC exposure led to differential gene expression in the skeletal muscle at both the infant and fetal developmental stages, albeit at varying significance. THC exposure during the fetal developmental period led to limited changes in gene expression, while the infant developmental period had more pronounced differences. This suggests that prenatal THC exposure has a progressive effect on differential gene expression. The PCA of gene expression data showed marked overlap between the control and THC-exposed groups for the fetal cohort, while the overlap in the infant cohort was minimal.

Activation of the ECS by THC led to differential expression of genes involved in the PI3K-Akt signaling pathway in the fetal cohort. The PI3K-Akt signaling pathway plays an important role in cell metabolism, growth, and proliferation [42]. Of these cellular processes, the DEG are most involved in cell survival and apoptosis. Alteration of the PI3K-Akt signaling pathway gene expression in skeletal muscle by THC exposure is in accordance with previously published work that showed acute administration of THC led to activation of the PI3K-Akt-GSK-3 signaling pathway mediated by activation of CB1R in the brain. The authors speculated that activation of the PI3k-Akt-GSK-3 signaling pathway may be related to the neuroprotective effects of ECS activation [43]. In the context of skeletal muscle, activation of the PI3K-Akt signaling pathway has been shown to induce skeletal muscle hypertrophy (increase in skeletal muscle mass) in a manner dependent on upstream insulin-like growth factor 1 (IGF-1) activity [44]. In the present study, the low number of DEG and the fact that IGF-1 was not upregulated in the skeletal muscle suggests that the alteration in protein expression related to PI3K-Akt signaling likely does not translate to functional changes in skeletal muscle mass.

In addition to PI3k-Akt signaling, activation of the ECS through THC exposure during early development led to changes in expression of genes involved in inflammation, specifically interleukin (IL) signaling. IL-4 and IL-13 are key mediators of allergic inflammation. They contribute to changing the immunoglobulin class of both IgE and IgG4, as well as the activation of eosinophils, basophils, and mast cells [45]. IL-10 has potent anti-inflammatory effects, limiting host response to pathogens [46], while IL-24 is a modulator of the immune response and plays a regulatory role in tissue homeostasis, host defense, and oncogenesis [47]. In the fetal cohort, 3 out of 4 of the genes involved in IL-24 signaling were upregulated. In the infant cohort, 6 out of 8 genes involved in IL-24 signaling were upregulated and all 7 genes involved in IL-4, IL-13, and IL-10 signaling were upregulated. These results suggest that activation of the ECS through THC in skeletal muscle may alter cytokine signaling, having both pro- and anti-inflammatory effects. Dysregulation of inflammatory signaling in the skeletal muscle is noteworthy considering a precise balance of pro- and anti-inflammatory cytokines are critical in skeletal muscle homeostasis, myogenesis, and response to injury [48–50].

While activation of the ECS by THC in the skeletal muscle led to changes in gene expression, the limited number of genes suggest that THC exposure during development plays a negligible role in skeletal muscle development. However, it remains unclear if more sensitive assays would have detected differences in skeletal muscle development caused by THC exposure. This study utilized FFPE tissues, despite known challenges with RNA quality. We strategically employed Nanostring nCounter technology, a method specifically optimized to extract

high-quality gene expression data from FFPE tissues. While this approach was effective, it's possible that some DEG may not have been detected. Future studies utilizing fresh frozen tissues and RNA sequencing could further enhance our understanding of THC-induced differential gene expression in skeletal muscle.

Although prenatal THC exposure seemingly has limited effects on skeletal muscle development, previous studies have shown prenatal THC use is associated with increased risk of stillbirth, low birth weight for gestational age, preterm delivery, and offspring neurodevelopmental effects [51–55]. These associations are important to consider as synthetic CB1 and CB2 ligands are being developed to modulate the ECS in health and disease [56]. Although therapeutic cannabinoids have shown promise, the associated psychoactive effects have limited their use clinically. Thus, motivating the search for natural and synthetic cannabinoids that maintain medicinal properties but lack psychotropic effects. To date, the FDA has approved one cannabis-derived drug product: Epidiolex (cannabidiol) and 3 synthetic cannabis-related drug products: Marinol (dronabinol), Syndros (dronabinol), and Cesamet (nabilone). With the continued development of CB1 and CB2 ligands, testing for safety in pregnant individuals, and the potential effects on the developing fetus, become increasingly important.

The present study suggests that prenatal THC exposure leads to limited changes in differential gene expression in the skeletal muscle of fetal and infant rhesus macaques. The most prominent change was an increase in pro-inflammatory signaling, which suggests the potential for tissue damage and atrophy. However, future studies are needed to assess whether these changes in gene expression are associated with adverse functional outcomes in the skeletal muscle. These findings contribute to the limited data on the safety of prenatal THC exposure and shed light on pathways impacted by increased ECS signaling during development.

## Acknowledgments

We would like to thank the NIDA Drug Supply Program in addition to the veterinary and husbandry staff at ONPRC who provided excellent care for the animals used in this study. In particular, we would like to thank Dr. Lauren Drew Martin, Dr. Heather Sidener, Travis Hodge, and Trent Crowley. Additionally, we would like to thank ONPRC Pathology Services Unit and the ONPRC Integrated Pathology Core.

## Author Contributions

**Conceptualization:** Samantha A. Moellmer, Jamie O. Lo, Karina H. Nakayama.

**Data curation:** Samantha A. Moellmer, Olivia L. Hagen, Parsa A. Farhang, Victoria R. Duke, Jamie O. Lo, Karina H. Nakayama.

**Formal analysis:** Samantha A. Moellmer, Victoria R. Duke, Jamie O. Lo, Karina H. Nakayama.

**Funding acquisition:** Jamie O. Lo, Karina H. Nakayama.

**Investigation:** Jamie O. Lo.

**Methodology:** Samantha A. Moellmer, Olivia L. Hagen, Parsa A. Farhang, Meghan E. Fallon, Jamie O. Lo, Karina H. Nakayama.

**Resources:** Jamie O. Lo, Karina H. Nakayama.

**Supervision:** Owen J. T. McCarty, Jamie O. Lo, Karina H. Nakayama.

**Validation:** Samantha A. Moellmer.

**Visualization:** Samantha A. Moellmer.

**Writing – original draft:** Samantha A. Moellmer, Olivia L. Hagen, Parsa A. Farhang, Victoria R. Duke, Meghan E. Fallon, Monica T. Hinds, Owen J. T. McCarty, Jamie O. Lo, Karina H. Nakayama.

**Writing – review & editing:** Samantha A. Moellmer, Olivia L. Hagen, Parsa A. Farhang, Victoria R. Duke, Meghan E. Fallon, Monica T. Hinds, Owen J. T. McCarty, Jamie O. Lo, Karina H. Nakayama.

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
