## [Decision Letter · Decision Letter 0]

2 May 2024

PONE-D-24-10539Effects of in utero delta-9-tetrahydrocannabinol (THC) exposure on fetal and infant musculoskeletal developmentPLOS ONE

Dear Dr. Nakayama,

Thank you for submitting your manuscript to PLOS ONE. After careful consideration, we feel that it has merit but does not fully meet PLOS ONE’s publication criteria as it currently stands. Therefore, we invite you to submit a revised version of the manuscript that addresses the points raised during the review process.

We look forward to receiving your revised manuscript.

Kind regards,

Vinod K. Yaragudri, MS, Ph.D

Academic Editor

PLOS ONE

Journal Requirements:

2. In order to comply with PLOS ONE's guidelines for non-human primate experiments (http://journals.plos.org/plosone/s/submission-guidelines#loc-non-human-primates), please provide additional details regarding housing conditions, feeding regimens, environmental enrichment, and all relevant steps taken to alleviate suffering (anaesthesia, analgesia, details about humane endpoints, euthanasia, etc.). Also indicate how often animal care staff monitored the health and well-being of the animals and the criteria used to make such assessments. Lastly, specify the disposition of animals at the end of the study (e.g. euthanasia, returned to home colony, etc.), please provide this information for both mothers and infants involved in the study. If animals were euthanized following the study, please provide the method of sacrifice.

Reviewers' comments:

Reviewer's Responses to Questions

**Comments to the Author**

1. Is the manuscript technically sound, and do the data support the conclusions?

Reviewer #1: Partly

Reviewer #2: Yes

2. Has the statistical analysis been performed appropriately and rigorously? 

Reviewer #1: Yes

Reviewer #2: Yes

3. Have the authors made all data underlying the findings in their manuscript fully available?

Reviewer #1: Yes

Reviewer #2: Yes

4. Is the manuscript presented in an intelligible fashion and written in standard English?

Reviewer #1: Yes

Reviewer #2: Yes

5. Review Comments to the Author

Reviewer #1: This is a manuscript where the authors have the authors have assessed gene expression changes in skeletal muscle from fetal and infant rhesus monkeys that were exposed to delta-9-THC (THC) during gestation.

The authors should clarify where in the central nervous system that CB2R is expressed. Most cannabinoid researchers do not believe that CB2R is expressed in neurons under normal, non-injury conditions.

It is not accurate that THC is an antagonist of CB2R.

The listed dosages are expressed in a non-standard manner. It would be better to express dosage of THC as mg/kg/day for comparison to other studies.

It is not clear that animals consumed all of the cookies (containing THC). Was cookie (and THC) consumption confirmed and verified?

The A260/A280, RIN, and DV200 values are very low. While I understand that these are comparable for work done using FFPE tissues, it does raise the possibility that some differentiated expressed genes (DEGs) were missed due to low RNA quality? This limitation should be mentioned in the discussion as well as the point that future work should/could address this question using fresh frozen tissue.

In lines 196-198, the authors state that prenatal THC exposure leads to epigenetic gene expression changes. While this might be true, it seems to suggest that the DEGs observed in the current study were due to epigenetic changes, something for which there is no evidence for in the current work.

Although there weren't glaring problems with the current manuscript, it also doesn't seem to represent a complete story. This pilot study would be much stronger and more compelling if there was more to the story.

Reviewer #2: The endocannabinoid system (ECS) orchestrates crucial bodily functions and responses to external factors. While explored as a therapeutic target, understanding ECS modulation across tissues is essential. The team investigated the impacts of experimental exposure to natural plant THC (exogenous; a robust ligand of cannabinoid receptors) during pregnancy, on fetal and infant muscle development using a non-human primate model. The authors’ findings highlight subtle effects on muscle development, with notable changes in inflammation and cytokine signaling pathways, indicating potential tissue damage.

Authors start with the premise that any therapeutic modulation of ECS by THC may heavily depend on patho/physiological state, based on studies in rodents. Specifically, considering that activation of CB1R through EC 2-AG can disrupt myoblast differentiation, the authors ask if enhanced activation via exposure to exogenous cannabinoids might affect early skeletal muscle development in primates. The team has used a chronic regimen using escalating oral dosing (constituted in chow/edibles) of 0-2.5 mg/7kg body weight per day over 11-12 weeks (plus maintenance with the highest THC dose) from conception until sample collection from, in time-mated, timed pregnant rhesus macaques. Authors sought to characterize (the impact of) neuroinflammatory response on fetal and infant stages of skeletal muscle development, evaluated by histomorphology and for differential gene expression using a Nanostring nCounter neuroinflammatory panel, to gain insights to shed light on key pathways impacted by THC exposure-induced stimulation of the ECS during early embryonic, fetal development.

This important study, lays the groundwork for further exploration into the long-term, intergenerational consequences of prenatal THC exposure on neuroinflammatory profiles and functional outcomes in primates, and humans.

Comments: Very important piece of work in the area. All aspects of the study design, data analysis, inference, presentation, literature citations are adequate. Methodology is rigorous and meet the requirements of PLOS One.

6. PLOS authors have the option to publish the peer review history of their article (what does this mean?). If published, this will include your full peer review and any attached files.

Reviewer #1: No

Reviewer #2: No

---

## [Author Response · Author response to Decision Letter 0]

14 May 2024

We thank the reviewers for their insightful comments and suggestions to improve the quality of our manuscript (PONE-D-24-10539) “Effects of in utero delta-9-tetrahydrocannabinol (THC) exposure on fetal and infant musculoskeletal development”. Based on reviewer feedback, we have improved the manuscript by clarifying background information and expanded on future work that can be done to progress the field. We are happy to submit the current revised version and hope to meet the high-quality standards of your journal.

Please find below our point-by-point responses to reviewer comments below: 

Journal requirements

1. In order to comply with PLOS ONE's guidelines for non-human primate experiments (http://journals.plos.org/plosone/s/submission-guidelines#loc-non-human-primates), please provide additional details regarding housing conditions, feeding regimens, environmental enrichment, and all relevant steps taken to alleviate suffering (anaesthesia, analgesia, details about humane endpoints, euthanasia, etc.). Also indicate how often animal care staff monitored the health and well-being of the animals and the criteria used to make such assessments. Lastly, specify the disposition of animals at the end of the study (e.g. euthanasia, returned to home colony, etc.), please provide this information for both mothers and infants involved in the study. If animals were euthanized following the study, please provide the method of sacrifice.

Thank you for highlighting this, we have provided additional details in the Methods section regarding housing conditions, feeding regimens, steps to alleviate suffering, monitoring of animal welfare, and disposition of animals.

Here is the excerpt:

“Animals were observed at least twice a day by trained animal care technicians and every effort is made by both veterinary and research staff to minimize pain and distress in the rhesus macaques as clinically needed. During the course of the study, clinical signs of distress in the pregnant dams such as weight loss, anorexia, cachexia, self-injury, or failure to thrive were assessed by ONPRC Division of Comparative staff and the ONPRC Behavioral Management Program to ensure psychological health and well-being. When possible, positive reinforcement training was used to train rhesus macaques to cooperate with various procedures, thus reducing the stress associated with those procedures. Rhesus macaques were housed in full contact pairs with a compatible social partner for the duration of the study in accordance with the Guide for the Care and Use of Laboratory Animals”

“This study used indoor-housed adult, female rhesus macaques (n=18) of similar size (~6-7 kg) maintained on a standard chow diet (TestDiet, St. Louis, Missouri) given twice a day in addition to fresh produce or other food enrichment daily, with and tap water available ad libitum”

“All methods of euthanasia utilized at the ONPRC are consistent with the American Veterinary Medical Association Guidelines for the Euthanasia of Animals , sodium pentobarbital was delivered intravenously followed by exsanguination under a deep surgical plane of anesthesia.”

Thank you for noticing that discrepancy. We will be sure to provide the correct grant numbers for the awards upon resubmission.

The Nanostring nCounter raw and normalized data has been made available on Kaggle, doi: 10.34740/kaggle/dsv/7853952. We will add this information into our data availability statement.

Reviewer #1

This is a manuscript where the authors have the authors have assessed gene expression changes in skeletal muscle from fetal and infant rhesus monkeys that were exposed to delta-9-THC (THC) during gestation.

Specific comments follow:

1. The authors should clarify where in the central nervous system that CB2R is expressed. Most cannabinoid researchers do not believe that CB2R is expressed in neurons under normal, non-injury conditions.

Thank you for this comment, we have made clarifications regarding where in the central nervous system CB2R is expressed in the introduction.

Here is the excerpt:

“There is conflicting evidence surrounding CB2R expression in the central nervous system, with some studies suggesting CB2R is weakly expressed in hippocampal and brain stem neurons under normal physiological conditions [2, 3]. However, under inflammatory conditions or injury, CB2R is highly expressed on reactive microglia [4, 5]. Outside of the nervous system, CB2R is highly expressed in peripheral immunological tissues [4].”

2. It is not accurate that THC is an antagonist of CB2R.

Thank you to the reviewer for pointing out this discrepancy. We have made this clarification in the introduction. 

Here is the excerpt:

“THC is the main psychoactive component in cannabis and acts as a partial agonist on CB1R and CB2R [12, 13]. In addition to its role as a partial agonist on the CB2R receptor, THC can inhibit the activity of cannabinoid ligands on CB2R, thereby also acting as a weak antagonist in select settings [14].”

3. The listed dosages are expressed in a non-standard manner. It would be better to express dosage of THC as mg/kg/day for comparison to other studies.

Thank you for this suggestion, we have made this correction throughout the manuscript.

4. It is not clear that animals consumed all of the cookies (containing THC). Was cookie (and THC) consumption confirmed and verified?

Yes, cookies were administered in the morning prior to daily chow and complete ingestion was confirmed. These details were added into the methods.

Here is the excerpt:

“Cookies were administered in the morning prior to daily chow to ensure complete ingestion.”

5. The A260/A280, RIN, and DV200 values are very low. While I understand that these are comparable for work done using FFPE tissues, it does raise the possibility that some differentiated expressed genes (DEGs) were missed due to low RNA quality? This limitation should be mentioned in the discussion as well as the point that future work should/could address this question using fresh frozen tissue.

This is an excellent point, and we have now included additional information outlining this in the discussion.

Here is the excerpt:

“This study utilized FFPE tissues, despite known challenges with RNA quality. We strategically employed Nanostring nCounter technology, a method specifically optimized to extract high-quality gene expression data from FFPE tissues. While this approach was effective, it’s possible that some differentially expressed genes (DEGs) may not have been detected. Future studies utilizing fresh frozen tissues and RNA sequencing could further enhance our understanding of THC-induced differential gene expression in skeletal muscle.”

6. In lines 196-198, the authors state that prenatal THC exposure leads to epigenetic gene expression changes. While this might be true, it seems to suggest that the DEGs observed in the current study were due to epigenetic changes, something for which there is no evidence for in the current work.

Thank you for this comment. We agree that discussing epigenetic changes may be misleading to readers in suggesting the DEGs are caused by epigenetic changes. We have decided to remove details of THC and epigenetic changes.

7. Although there weren't glaring problems with the current manuscript, it also doesn't seem to represent a complete story. This pilot study would be much stronger and more compelling if there was more to the story.

We appreciate this feedback. This pilot study lays the groundwork for future exploration into the effects of prenatal THC-exposure on musculoskeletal development. As per your recommendation, we have added a limitations and future directions section into the discussion.

Here is the excerpt:

“This study utilized FFPE tissues, despite known challenges with RNA quality. We strategically employed Nanostring nCounter technology, a method specifically optimized to extract high-quality gene expression data from FFPE tissues. While this approach was effective, it’s possible that some differentially expressed genes (DEGs) may not have been detected. Future studies utilizing fresh frozen tissues and RNA sequencing could further enhance our understanding of THC-induced differential gene expression in skeletal muscle.”

Reviewer #2

The endocannabinoid system (ECS) orchestrates crucial bodily functions and responses to external factors. While explored as a therapeutic target, understanding ECS modulation across tissues is essential. The team investigated the impacts of experimental exposure to natural plant THC (exogenous; a robust ligand of cannabinoid receptors) during pregnancy, on fetal and infant muscle development using a non-human primate model. The authors’ findings highlight subtle effects on muscle development, with notable changes in inflammation and cytokine signaling pathways, indicating potential tissue damage.

Authors start with the premise that any therapeutic modulation of ECS by THC may heavily depend on patho/physiological state, based on studies in rodents. Specifically, considering that activation of CB1R through EC 2-AG can disrupt myoblast differentiation, the authors ask if enhanced activation via exposure to exogenous cannabinoids might affect early skeletal muscle development in primates. The team has used a chronic regimen using escalating oral dosing (constituted in chow/edibles) of 0-2.5 mg/7kg body weight per day over 11-12 weeks (plus maintenance with the highest THC dose) from conception until sample collection from, in time-mated, timed pregnant rhesus macaques. Authors sought to characterize (the impact of) neuroinflammatory response on fetal and infant stages of skeletal muscle development, evaluated by histomorphology and for differential gene expression using a Nanostring nCounter neuroinflammatory panel, to gain insights to shed light on key pathways impacted by THC exposure-induced stimulation of the ECS during early embryonic, fetal development.

This important study, lays the groundwork for further exploration into the long-term, intergenerational consequences of prenatal THC exposure on neuroinflammatory profiles and functional outcomes in primates, and humans.

Comments: Very important piece of work in the area. All aspects of the study design, data analysis, inference, presentation, literature citations are adequate. Methodology is rigorous and meet the requirements of PLOS One.

Thank you for these comments and summary.

---

## [Decision Letter · Decision Letter 1]

25 Jun 2024

Effects of in utero delta-9-tetrahydrocannabinol (THC) exposure on fetal and infant musculoskeletal development

PONE-D-24-10539R1

Dear Dr. Nakayama,

We’re pleased to inform you that your manuscript has been judged scientifically suitable for publication and will be formally accepted for publication once it meets all outstanding technical requirements.

Kind regards,

Vinod K. Yaragudri, MS, Ph.D

Academic Editor

PLOS ONE

Additional Editor Comments (optional):

**Editorial Comment:** Please modify the title of your manuscript to indicate the species in which the study was performed. For example please add "preclinical nonhuman primate model" or "in rhesus macaques" to the title.

Reviewers' comments:

Reviewer's Responses to Questions

**Comments to the Author**

1. If the authors have adequately addressed your comments raised in a previous round of review and you feel that this manuscript is now acceptable for publication, you may indicate that here to bypass the “Comments to the Author” section, enter your conflict of interest statement in the “Confidential to Editor” section, and submit your "Accept" recommendation.

Reviewer #1: All comments have been addressed

Reviewer #2: All comments have been addressed

2. Is the manuscript technically sound, and do the data support the conclusions?

Reviewer #1: Yes

Reviewer #2: Yes

3. Has the statistical analysis been performed appropriately and rigorously? 

Reviewer #1: Yes

Reviewer #2: Yes

4. Have the authors made all data underlying the findings in their manuscript fully available?

Reviewer #1: Yes

Reviewer #2: Yes

5. Is the manuscript presented in an intelligible fashion and written in standard English?

Reviewer #1: Yes

Reviewer #2: Yes

6. Review Comments to the Author

Reviewer #1: (No Response)

Reviewer #2: (No Response)

7. PLOS authors have the option to publish the peer review history of their article (what does this mean?). If published, this will include your full peer review and any attached files.

Reviewer #1: No

Reviewer #2: No

---

## [Editor Report · Acceptance letter]

9 Jul 2024

PONE-D-24-10539R1 

PLOS ONE

Dear Dr. Nakayama, 

I'm pleased to inform you that your manuscript has been deemed suitable for publication in PLOS ONE. Congratulations! Your manuscript is now being handed over to our production team.

Kind regards, 

on behalf of

Dr. Vinod K. Yaragudri 

Academic Editor

PLOS ONE